# Functional Characterization of 21 Rare Allelic *CYP1A2* Variants Identified in a Population of 4773 Japanese Individuals by Assessing Phenacetin *O*-Deethylation

**DOI:** 10.3390/jpm11080690

**Published:** 2021-07-22

**Authors:** Masaki Kumondai, Evelyn Marie Gutiérrez Rico, Eiji Hishinuma, Yuya Nakanishi, Shuki Yamazaki, Akiko Ueda, Sakae Saito, Shu Tadaka, Kengo Kinoshita, Daisuke Saigusa, Tomoki Nakayoshi, Akifumi Oda, Noriyasu Hirasawa, Masahiro Hiratsuka

**Affiliations:** 1Laboratory of Pharmacotherapy of Life-Style Related Diseases, Graduate School of Pharmaceutical Sciences, Tohoku University, Sendai 980-8578, Japan; masaki.kumondai.d5@tohoku.ac.jp (M.K.); gutierrez@tohoku.ac.jp (E.M.G.R.); yuynakan@ncc.go.jp (Y.N.); shuki.yamazaki.q7@dc.tohoku.ac.jp (S.Y.); noriyasu.hirasawa.c7@tohoku.ac.jp (N.H.); 2Department of Pharmaceutical Sciences, Tohoku University Hospital, Sendai 980-8574, Japan; 3Advanced Research Center for Innovations in Next-Generation Medicine, Tohoku University, Sendai 980-8575, Japan; ehishi@ingem.oas.tohoku.ac.jp (E.H.); akiko.ueda.d1@tohoku.ac.jp (A.U.); ssaito@megabank.tohoku.ac.jp (S.S.); 4Tohoku Medical Megabank Organization, Tohoku University, Sendai 980-8575, Japan; tadaka@megabank.tohoku.ac.jp (S.T.); kengo@ecei.tohoku.ac.jp (K.K.); saigusa@tohoku.ac.jp (D.S.); 5Faculty of Pharmacy, Meijo University, Nagoya 468-8503, Japan; nakayoshi@hiroshima-cu.ac.jp (T.N.); oda@meijo-u.ac.jp (A.O.)

**Keywords:** cytochrome P450 1A2, genetic variation, phenacetin, drug metabolism

## Abstract

Cytochrome P450 1A2 (CYP1A2), which accounts for approximately 13% of the total hepatic cytochrome content, catalyzes the metabolic reactions of approximately 9% of frequently used drugs, including theophylline and olanzapine. Substantial inter-individual differences in enzymatic activity have been observed among patients, which could be caused by genetic polymorphisms. Therefore, we functionally characterized 21 novel CYP1A2 variants identified in 4773 Japanese individuals by determining the kinetic parameters of phenacetin *O*-deethylation. Our results showed that most of the evaluated variants exhibited decreased or no enzymatic activity, which may be attributed to potential structural alterations. Notably, the Leu98Gln, Gly233Arg, Ser380del Gly454Asp, and Arg457Trp variants did not exhibit quantifiable enzymatic activity. Additionally, three-dimensional (3D) docking analyses were performed to further understand the underlying mechanisms behind variant pharmacokinetics. Our data further suggest that despite mutations occurring on the protein surface, accumulating interactions could result in the impairment of protein function through the destabilization of binding regions and changes in protein folding. Therefore, our findings provide additional information regarding rare CYP1A2 genetic variants and how their underlying effects could clarify discrepancies noted in previous phenotypical studies. This would allow the improvement of personalized therapeutics and highlight the importance of identifying and characterizing rare variants.

## 1. Introduction

The individual impact and clinical significance of genetic factors on drug metabolism have been well established in vivo and in vitro [1,2,3,4,5]. Several gene-coding alleles for drug-metabolizing enzymes and transporters correspond to distinct phenotypes [6,7], making it necessary to account for distinctive environmental factors and inter-individual genetic differences while establishing rational prescribing decisions to ensure an optimal pharmacological response.

The drug-metabolizing enzyme cytochrome P450 1A2 (CYP1A2) is constitutively expressed in the human liver and accounts for approximately 13% of the total hepatic cytochrome (CYP) content [8,9]. It catalyzes the metabolism of approximately 9% of commonly used drugs, including the oxidation of drugs such as theophylline, olanzapine, and propranolol [10,11]. CYP1A2 plays an important role in the metabolism of psychotropic drugs and other drugs targeting the central nervous system, either involved in the catalysis of the main metabolic route or directly inhibited by these drugs. Previous studies have identified a 40- to 130-fold inter-individual difference in CYP1A2 activity and a 40-fold difference in protein expression among individuals; of these, 35–75% of differences are attributed to genetic factors [12]. To date, 40 polymorphic variants of the CYP1A2 gene have been reported (https://www.pharmvar.org/gene/CYP1A2 accessed on 21 July 2021), of which the most frequent and most highly characterized variant is the -163C>A polymorphism (*CYP1A2*1F*), located in the promoter region. High frequencies of this variant have been reported in populations worldwide, including among the Japanese population (60%) [13,14]. To date, several studies have examined the effects of polymorphisms of the *CYP1A2* gene on enzyme inducibility. Examples include a report of decreased blood levels of olanzapine in schizophrenic patients with the -163C>A polymorphism, and one discussing increased clearance of theophylline in non-smoker asthmatic patients carrying the -3860C>A polymorphism [15,16].

In a continuous effort to improve medical outcomes, identify potential risks associated with treatments, and establish population-specific clinical tools (including dosing algorithms and adverse effect predictors), the Tohoku Medical Megabank Organization (ToMMo) reported whole-genome sequences of 4773 Japanese individuals (https://www.megabank.tohoku.ac.jp/english/timeline/20190913_01 accessed on 23 March 2021) up to September 2019. Twenty-one single-nucleotide variations were identified in the regions coding for CYP1A2. Amino acid substitutions may result in alterations in the CYP1A2 protein structure and consequently affect enzyme function; this, in turn, may significantly affect drug efficacy and increase the risk of adverse drug reactions. While the importance of evaluating the clinical impact of these variants in patients is paramount to predicting treatment outcomes, polymorphic variants of the *CYP1A2* gene causing protein structure alteration are rare within the Japanese population, with a frequency of approximately only 1% [17,18,19]. This, along with the increased patient risk and the potential for adverse effects specific to CYP1A2 substrates, makes it challenging to conduct clinical trials. Therefore, in vitro functional analysis of CYP1A2 variants has been widely performed [3,5,20,21,22], including our previous comprehensive analysis of the enzymatic function of 19 CYP1A2 variants. Such analyses make it possible to identify the causes of inter-individual variations in response to endogenous substances, and to devise strategies to identify patients at risk of therapeutic failure or adverse drug reactions.

This study aimed to comprehensively elucidate the in vitro functional changes of the 21 novel structural variants arising from *CYP1A2* gene polymorphisms, which were identified in 4773 Japanese individuals. We conducted recombinant protein expression in 293FT cells, with co-expression of cytochrome P450 oxidoreductase (CPR) and cytochrome b_5_. Characterization of enzymatic activity was performed using phenacetin as the preferred probe for the in vitro screening of CYP1A2-based metabolism [23,24,25]. To further analyze the mechanisms underlying these differences, carbon monoxide (CO)-difference spectroscopy and three-dimensional (3D) docking analyses were performed.

## 2. Materials and Methods

### 2.1. Chemicals

Reagents used in this study were purchased from the following sources: phenacetin (Nacalai Tesque, Kyoto, Japan); 4-acetamidophenol (Sigma-Aldrich, St. Louis, MO, USA); 4-acetamidophenol-d4 (Cayman Chemical, Ann Arbor, MI, USA); oxidized β-nicotinamide-adenine dinucleotide phosphate (NADP^+^), glucose-6-phosphate (G-6-P), glucose-6-phosphate dehydrogenase (G-6-PDH), reduced β-nicotinamide-adenine dinucleotide (NADH), oxidized β-nicotinamide-adenine dinucleotide, and reduced β-nicotinamide-adenine dinucleotide phosphate (NADPH) (Oriental Yeast, Tokyo, Japan); polyclonal anti-human CYP1A2 antibody (cat. no. ab170204; Abcam, Cambridge, UK); horseradish peroxidase (HRP)-conjugated goat anti-rabbit IgG (ProteinSimple, Tokyo, Japan); and sodium cyanide and cytochrome c from horse hearts (Nacalai Tesque, Kyoto, Japan). All other chemicals and reagents were of the highest quality and commercially available.

### 2.2. Sanger Sequencing Analysis for the Detection of CYP1A2 Sequence Alterations

Peripheral blood leukocytes were isolated from the whole blood of participating Japanese subjects of the community-based cohort study conducted by ToMMo. Written informed consent was obtained from all subjects prior to sample collection [26]. Polymerase chain reaction (PCR) amplification of the genomic DNA extracted from the cells was conducted using a Gentra Puregene Blood Kit (Qiagen, Hilden, Germany). More than 10 ng genomic DNA, 2 × AmpliTaq Gold 360 Master Mix (Applied Biosystems, Foster City, CA, USA), and 0.5 µM of each primer (Appendix A), in a total volume of 20 µL were used for PCR amplification. The thermal cycling conditions included an initial denaturation at 95 °C for 10 min, followed by 30 cycles of denaturation at 95 °C for 30 s, annealing at 60 °C for 30 s, extension at 72 °C for 1 min (exons 2, 4 and 5) or 30 s (exons 3, 6, and 7), and a final extension at 72 °C for 7 min. The PCR products were purified on a column and analyzed by Sanger sequencing using the same primers for each exon as employed for PCR.

### 2.3. Expression of CYP1A2 Variants in 293FT Cells

Wild-type and variant CYP1A2 cDNAs subcloned into the pcDNA3.4 vector were purchased from GenScript (Piscataway, NJ, USA). Plasmids carrying CPR or cytochrome b_5_ cDNAs subcloned into the pcDNA3.4 vector were prepared as previously described [27]. 293FT cells (ThermoFisher Scientific, Waltham, MA, USA) were cultured in Dulbecco’s modified Eagle’s medium (Nacalai Tesque) containing 10% fetal bovine serum at 37 °C under 5% CO_2_. Microsomal fractions containing each CYP1A2 variant, CPR, and cytochrome b_5_ were prepared according to the previously described protocol [27]. Protein concentrations were determined using a BCA Protein Assay Kit (ThermoFisher Scientifics).

### 2.4. Western Blotting

Immunoassays were performed using the Wes (ProteinSimple, San Jose, CA, USA) and Compass for SW ver. 4.1.0 (ProteinSimple) software to evaluate CYP1A2 protein expression levels. Briefly, 100 ng of microsomes containing CYP1A2 wild-type and variant proteins were loaded into each well. CYP1A2 levels were detected using a polyclonal anti-CYP1A2 antibody (1:100 dilution) and HRP-conjugated goat anti-rabbit IgG. A total protein assay was performed to normalize each signal using 100 ng of microsomes, as per the manufacturer’s instructions [28]. 

### 2.5. Determination of Cytochrome (CYP), Cytochrome P450 Oxidoreductase (CPR), and Cytochrome b_5_ Content

CYP1A2 holoprotein, CPR, and cytochrome b_5_ contents were spectrophotometrically measured by ultraviolet-visible spectrophotometry (Cary 300 UV-Vis spectrophotometer, Agilent Technologies, Santa Clara, CA, USA), as previously reported [27,29]. Data analysis was conducted using Jasco Spectra Manager (JASCO Corporation, Sendai, Japan). Cuvettes (Sub-Micro Cells; 16.50-Q-10/Z20) were purchased from Starna Scientific, Ltd. (London, UK).

### 2.6. Phenacetin O-Deethylation

The extent of *O*-deethylation of phenacetin by CYP1A2 was measured using a modification of the method previously reported by Ito et al. [21,27]. The reaction mixture (150 μL) was composed of the microsomal fraction (25 μg), phenacetin (2.5, 5, 10, 25, 50, 100, 250, or 500 μM), 3.3 mM MgCl_2_, and 50 mM potassium phosphate buffer (pH 7.4). After pre-incubating at 37 °C for 3 min, reactions were initiated by adding the NADPH-generation system, consisting of 1.3 mM NADP^+^, 3.3 mM G-6-P, and 0.4 U/mL G-6-PDH. The mixture was incubated at 37 °C for 40 min, before terminating the reactions by the addition of 150 μL acetonitrile, containing 5 μM acetamidophenol-d4 as an internal standard. After protein removal by centrifugation at 15,400× *g* for 10 min, 5 μL of the supernatant was injected into a liquid chromatography-tandem mass spectrometry system, as previously described [27].

### 2.7. 3D Structural Modeling of CYP1A2

The 3D structural modeling of CYP1A2 was conducted based on the X-ray structure of CYP1A2 reported by Sansen et al. (Protein Data Bank code: 2HI4) [30]. Phenacetin was coordinated with the CYP1A2 wild-type model structure using the CDOCKER protocol of Discovery Studio 2.5 (BIOVIA, San Diego, CA, USA). Docking iterations were conducted considering the binding orientations and binding energies under the conditions previously described by Oda et al. [31]. After each substitution, structural optimization was performed as previously reported [32]. Alongside 3D docking analysis, Polymorphism Phenotyping V-2 (PolyPhen-2) [33] and Sorting Intolerant From Tolerant (SIFT) [34] predictive software were used to evaluate the structural impact of the amino acid substitutions in CYP1A2 variants on the functionality of the enzyme [35].

### 2.8. Data Analysis

Kinetic parameters (*K_m_*; Michaelis constant, *V_max_*; maximum velocity, and *CL_int_*; intrinsic clearance) were determined using the Enzyme Kinetics Module of SigmaPlot 12.5 (Systat Software, Inc., Chicago, IL, USA), a curve-fitting program based on non-linear regression analysis. All values are expressed as the mean ± standard deviation (SD) of experiments performed in triplicate. Statistical analyses for multiple comparisons were performed through variance analysis by Dunnett’s T3 test or the Kruskal–Wallis method, using IBM SPSS Statistics Ver. 22 (International Business Machines, Armonk, NY, USA). The normality of our datasets was initially assessed using the Shapiro-Wilk test. Differences were considered statistically significant when *P* values were less than 0.05.

## 3. Results

Twenty-one novel structural variants of CYP1A2 were identified in a cohort of 4773 Japanese individuals by whole-genome sequencing (allele frequencies ranging from 0.01–0.10%) and confirmed by Sanger sequencing, using the primer pairs listed in Appendix A. The structure-based enzymatic functionalities of all the variants were predicted using the PolyPhen-2 and SIFT software. PolyPhen-2 analysis revealed a total of ten potentially damaging substitutions, while SIFT analysis classified 12 variants as potentially damaging based on the effects of the detected amino acid substitutions on protein structure (Table 1).

### 3.1. Effect of CYP1A2 Variants on the Expression of CPR and Cytochrome b_5_

A previously established heterologous expression system was used to evaluate the structural and potential pharmacological properties of the novel CYP1A2 variants. In this technique, the variant proteins were overexpressed alongside CPR and cytochrome b_5_ in human embryonic kidney-derived 293FT cells. Wild-type and CYP1A2 variant microsomal fractions were isolated by differential centrifugation and western blotting using polyclonal CYP1A2 antibodies. All CYP1A2 variants were efficiently recognized at approximately 54 kDa for the quantification of expression levels. The quantified amounts were normalized to allow the determination of total protein concentrations (Figure 1). The CPR content of the variants did not differ significantly from that of wild-type, while the cytochrome b_5_ content of four variants (Arg34Trp, Gln344His, Ile351Thr, and His501Tyr) differed significantly from that of the wild-type (Table 2).

### 3.2. Holoprotein Content

The active holoprotein content of CYP1A2 wild-type and variant microsomal fractions was assessed by the spectrophotometric measurement of the increase in maximum absorption at 450 nm, following treatment with CO (Appendix A). The wild-type and 17 variants of CYP1A2 showed reduced CO spectra; among these, only the Leu491Met variant exhibited comparable holoenzyme content to that of the wild-type, while the remaining 16 variants (Arg34Trp, Glu44Lys, Gly47Asp, Arg79His, Asp104Tyr, Ala150Asp, Thr324Ile, Gln344His, Ile351Thr, Ile351Met, Arg356Gln, Gly454Asp, Arg457Trp, Val462Leu, Ile474Asn, and His501Tyr) exhibited significantly lower holoenzyme content. The holoenzyme content of the four remaining variants (Leu98Gln, Gly233Arg, Ser380del, and Ile401Thr) was found to be below the detectable limit for the method employed. The complete characteristics, including the content of the electron donors CPR and cytochrome b_5_ for all the CYP1A2 variant microsomes, are summarized in Table 2.

### 3.3. O-Deethylation of Phenacetin

Phenacetin *O*-deethylation activity was measured using a modification of previously described methods; kinetic parameters were subsequently determined using Michaelis–Menten curves (Figure 2). Enzymatic activities were normalized to the corresponding CYP1A2 total protein quantified by Western blotting (Table 3). The *O*-deethylation of phenacetin was assessed using microsomal protein fractions containing wild-type or variant CYP1A2 (25 μg); this revealed that 4-acetamidophenol was formed linearly, in a time-dependent manner, for incubation times of up to 40 min (data not shown). Among these, 13 variants (Glu44Lys, Gly47Asp, Arg79His, Asp104Tyr, Ala150Asp, Gly233Arg, Thr324Ile, Ile351Thr, Ile351Met, Arg356Gln, Ile401Thr, Val462Leu, Ile474Asn, Leu491Met, and His501Tyr) were found to show significantly lower *CL_int_* and *V_max_* values than the wild-type. Conversely, the *K_m_* values for all the variants evaluated did not differ significantly from those of the wild-type. Regarding Gln344His, this variant was found to have a *CL_int_* value slightly lower than that of the wild-type, while its *V_max_* and *K_m_* values did not differ significantly. The kinetic parameters of the remaining five variants (Leu98Gln, Gly233Arg, Ser380del, Gly454Asp, and Arg457Trp) could not be determined, since 4-acetamidophenol formation was negligible.

### 3.4. 3D Molecular Modeling

To evaluate the molecular properties of the identified novel variants, 3D structural modeling was performed; the structural integrity and molecular interactions within wild-type or CYP1A2 variant proteins were assessed when coordinated with phenacetin (Figure 3).

The Gly233Arg substitution introduced additional interactions with several residues, including additional hydrogen bonds with A’ helix members Val54 and Gly58; in turn, Gly58 formed additional interactions with Leu55, which resulted in the unraveling of a helical segment from Leu55 to Gly58. The lengthening of the carbon-hydrogen bond between the substituted Arg233 and Pro235 was also observed, resulting in interactions causing the G’ helix (Phe239 to Leu242) to unravel. Additional hydrogen bonds were formed between Asp320, Thr324, phenacetin, and heme in proximity to the substrate recognition site. This resulted in the loss of several interactions present with the wild-type enzyme, including interactions between Leu219 and Thr223; Asp320 and Gly316; π-alkyl interactions between phenacetin and heme; π-alkyl interactions between phenacetin and Ala317, and heme and Ala317; and π-π stacking interactions between heme, Phe451, and Phe381 (Figure 4A).

The substitution of the highly conserved Thr324 residue with Ile324 lead to the loss of the carbon-hydrogen bond between Ile324 and Thr321 and the two hydrogen bonds between the Ile324 and Asp320 residues. Meanwhile, additional π-alkyl interactions were introduced between Ile324 and Trp328, and Phe381 and Lys500, resulting in the loss of hydrophobic interactions between phenacetin and heme. New hydrogen bonds to Gly316 were also formed, while amide π-stacking interactions involving Gly316, Ala317, and phenacetin were lost (Figure 4B).

In the Ile401Thr variant, Thr401 formed an additional interaction with Thr394, but lost π-alkyl interactions with Leu86 and Cys405. Leu86 formed additional interactions with Val75 and Pro42. In the binding pocket, phenacetin did not show a hydrophobic interaction with Ala317, but did form an additional hydrogen bond with Gly316. A new interaction was introduced between heme and Ala325; however, the hydrophobic interactions between heme and Ile314 were lost. Additional interactions of Arg392 (attractive charges with Asp103 and carbon-hydrogen bond with Thr391) were observed, resulting in cascading interactions affecting Asp103 and Asp104, ultimately leading to the unwinding of the helical structure formed by Gly102–Lys106 (Figure 4C).

In the His501Tyr variant, Tyr501 lost several interactions with important structural and functional residues, including hydrogen bonds with the I helix residue Ser327 and the variable residue Ala502. Several π-σ and π-π T-shaped interactions with Trp328 were also lost, resulting in a change in the orientation of the alkyl interactions and the unraveling of the helix formed by residues Leu359 to Asp361. Additionally, phenacetin formed additional hydrogen bonds with Gly316, but lost two alkyl interactions between phenacetin and heme (Figure 4D).

## 4. Discussion

CYP1A2 is one of the major CYP enzymes in the human liver, forming 13% of the total hepatic CYP content [8,9]. CYP1A2 participates in the metabolic activation of dietary heterocyclic and aromatic amines, and has been shown to participate in the metabolic activation of procarcinogens along with CYP1A1 (another member of the CYP1A subfamily) [36,37]. Previous studies have linked the levels of expression of enzymes in this sub-family to the risk of carcinogenesis [38,39]. Considerable inter-individual differences are observed in the in vitro and in vivo enzymatic activity of CYP1A2. These differences become clinically relevant with respect to an individual’s response to CYP1A2-metabolized drugs [2,14,40,41]. Several studies have also addressed inter-individual variability through associative studies between the clearance of drugs metabolized by CYP1A2 and genetic polymorphisms [37,42,43,44]. However, the functional consequences of most CYP1A2 polymorphisms have not yet been determined, prompting the need for a detailed study of CYP1A2 variants, including rare, region-specific ones.

This study functionally characterized 21 CYP1A2 novel variants identified in a cohort of 4773 Japanese individuals. We heterologously expressed variants in an innovative mammalian expression system to measure CO spectra, CPR, and cytochrome b_5_ levels. We also conducted enzyme kinetic studies using phenacetin as a probe substrate, as well as in silico protein-ligand docking analysis using 3D structural models.

Analysis of the active holoprotein content revealed that only the Leu491Met variant exhibited holoenzyme content comparable to that of the wild-type. In contrast, 16 variants (Arg34Trp, Glu44Lys, Gly47Asp, Arg79His, Asp104Tyr, Ala150Asp, Thr324Ile, Gln344His, Ile351Thr, Ile351Met, Arg356Gln, Gly454Asp, Arg457Trp, Val462Leu, Ile474Asn, and His501Tyr) exhibited significantly lower holoenzyme contents. The holoenzyme content and enzyme activities of the four remaining variants (Leu98Gln, Gly233Arg, Ser380del, and Ile401Thr) could not be quantified. In silico 3D structural analysis of these four variants revealed differences in interactions of amino acid residues adjacent to the heme-binding site, which could explain their lack of detectable amounts of holoprotein content. Additional structural modifications observed at the substrate recognition site (SRS) and active site further supported the hypothesis that structural alterations could be responsible for the lack of enzymatic activity shown by these variants.

The Gly233Arg substitution, located in a highly conserved region, caused notable structural changes. The hydrophobic Gly residue often grants greater flexibility to the protein structure than Arg [45,46]. This substitution appears to cause changes in the proximity between the F helix and heme and Arg, allowing further stabilization of the protein structure and potentially altering its folding capacity. The uncoiling of the A’ and G’ helical segments (Leu55 to Gly58 and Phe239 to Leu424, respectively) may lower protein stability and affect the folding capacity and substrate binding ability through proximity alterations between the residues in the E and J helices; this could result in the tightening of the active site cavity (Figure 4A). Further changes alter the planar binding platform characteristic of CYP1A2 within the side chains of the F and I helixes; the Arg233 substitution causes the loss of alkyl interactions of both heme and the substrate with residues in the substrate-binding cavity (Thr223 and Asp320, Gly316, and Ala317), thus severing the Gly316-Ala317 peptide bond [47]. Further structural changes affecting heme include the loss of several stabilizing interactions, including alkyl interactions and π-π stacking interactions with Phe381 and the Cys-pocket residue Phe451. These structural changes could result in a highly unstable protein, which is incapable of properly binding heme and exhibits altered substrate docking or recognition.

Phenacetin *O*-deethylation activity studies revealed that of the 21 variants assayed, 13 (Glu44Lys, Gly47Asp, Arg79His, Asp104Tyr, Ala150Asp, Gly233Arg, Thr324Ile, Ile351Thr, Ile351Met, Arg356Gln, Ile401Thr, Val462Leu, Ile474Asn, Leu491Met, and His501Tyr) showed significantly decreased activity, resulting in the overall activity of the variants ranging from a slight decrease seen in Gln344His (80%), to the negligible enzymatic activity of Thr324 (1.45%), compared to wild-type. While four variants exhibited differences in cytochrome b_5_ content (Arf34Trp, Gln344His, Ile351Thr, and His501Tyr), this did not significantly impact their activities, as it is believed that CPR is predominantly responsible for electron transfer for CYP1A2 via an NADPH-dependent mechanism [48].

The highly conserved Thr324 residue, found within the I helix, is involved in the binding of oxygen to iron, and in the supply of protons required for enzyme activation. As a component of SRS-4, it acts as a proton donor for hydrogen bond formation [47,48,49]. The holoenzyme content of the Thr324Ile variant was found to be approximately 15% that of the wild-type. Consequently, this variant also showed the lowest clearance ratio among all the variants assayed in this study (1.45%). Ile is a hydrophobic residue, so substitution with Ile324 could cause the I helix to bend towards the core of the protein, reducing the active site cavity and resulting in the loss of interactions with Asp320 and Thr321 in SRS-4 (Figure 4B); these structural changes, along with the significantly increased *K_m_* and decreased *V_max_* values, suggest that this substitution may cause functional alterations that could impede proper substrate binding and recognition. Similarly, since CYP1A2 substrates bind above the heme group [48,50,51], additional interactions involving Trp328, Phe381, and Lys500 further impede proper substrate binding, as evidenced by the loss of stabilizing interactions between heme and phenacetin; this further supports our findings. Therefore, substitution with Ile324 may impair substrate binding and recognition, resulting in insufficient enzymatic activity. At the same time, heme stability is affected by further altering binding and stereoselectivity.

Western blotting results revealed that the Asp104Tyr variant showed significantly lower expression levels. The Tyr104 substitution, located in the B, B’ helix region, formed an additional hydrogen bond with Ile401 in the β 2-2 strand. As with the Arg79His substitution, this substitution also caused uncoiling from Gly102 to Lys106. These changes could lead to the misfolding of the resulting protein, leading to a decrease in the quantity of active protein, while possibly shifting substrate access channels; this would result in a decrease in the enzymatic activity without any significant effect on substrate affinity.

In the case of Ile401Thr, located in the 2-2 β-strand, the substitution of the hydrophobic Ile with the polar Thr caused substantial structural changes, resulting in a destabilized protein structure with altered enzymatic characteristics, as shown by its decreased enzymatic activity toward phenacetin (25.89% of the wild-type). Thr401 formed additional interactions with Thr401 and Thr394 in the 2-1 β sheet (Figure 4C), and differed in the stabilizing interactions with Val75, Leu86, and Cys405 of the 1-1, 1-2, and 1-3 β strands, respectively. It also formed additional alkyl interactions with Pro42 in the proline-rich region. Additionally, this substitution affected interactions with the substrate-binding residues, Gly316 and Ala317, and Ile314 (SRS-4), resulting in cascading interactions that lead to the unwinding of the Gly102–Lys106 helical structure, which flanks SRS-1. Modifying the highly conserved β sheets 1 and 2 could directly affect the hydrophobic substrate access channels, leading to poor substrate recognition [52,53]. Furthermore, CYP1A1 in silico studies have shown that although it is not directly involved in heme binding, the proline-rich region directly interacts with heme through the β-sheet and SRS segments, influencing holoenzyme formation [54,55]. While the holoenzyme content of the Ile401Thr variant could not be determined, it did exhibit enzymatic activity, albeit considerably reduced. The heme-bound fraction of this variant may have been formed by heme insertion early in the process of apoprotein folding [53]; while the formation may not have been substantial enough for holoenzyme to be detected using our methods, it may have been sufficient to exhibit minor enzymatic activity.

Gly454Asp and Arg457Trp, located in the heme-binding region, belong to a highly conserved ten-residue motif (FXXGXRXCXG) centered around Cys458, with Gly454 as the fourth, and Arg457 as the seventh base from the N-terminal side [56,57]. In the wild-type enzyme, Gly454 and Arg457 form a hydrogen bond, which was lost with the Asp454 substitution, while a carbon-hydrogen bond interaction with Leu98 was also lost, which in turn affected its interactions with Lys455. While the Trp457 substitution caused the loss of the helical turn from Met453 to Arg457. Substitutions in the 454th residue increased the distance between the metabolic iron and the phenacetin metabolic site from 4.32 Å in the wild-type, to 5.13 Å and 4.89 Å in the Asp454 and Trp457 variants, respectively; this indicates a significant change in the mobility of the surrounding area. The enzymatic activity of phenacetin was possibly abolished because of these changes. Moreover, the holoenzyme content for both variants was below 10% than that of the wild-type. A change in the proximity to the heme group and differences in regional flexibility could cause a change in the electrostatic interactions, affecting enzymatic activities and leading to inactive enzymes [58].

The His501Tyr variant showed reduced activity toward phenacetin (25.06%) compared to the wild-type. While the *K_m_* was higher than that of the wild-type, active holoenzyme content was significantly reduced, suggesting possible changes to the binding pocket. Ionic interactions between the residues may also be affected by the difference in pKa between His and Tyr [59]. Tyr501 lacked interactions with residues in the I helix, including the hydrogen bond with the highly conserved residue Ser327, and the π-σ and π-π T-shaped interactions with Trp328. Moreover, the alkyl interactions observed in this variant differed in their orientation and proximity to heme, leading to a change in substrate positioning, and possibly affecting substrate affinity. Likewise, the Tyr501 substitution led to cascading changes in the interactions, affecting Tyr189, Val220, Thr498, and Lys500 (members of a network of hydrogen-bonded water molecules and side chains) and Phe319. Phe319 appears to be involved in oxygen activation during catalysis, since mutations of this residue have been found to affect the coordination of iron and the kinetics of CO binding [30]. Therefore, even though this substitution occurs far from the center of the protein, accumulating modifications in the interaction could impair enzyme function by destabilizing key binding regions and affecting protein folding, as expected in the uncoiled region between Leu359 and Asp361.

In this study, 21 CYP1A2 variants were functionally characterized to gain a more comprehensive understanding of the structural effects of single amino acid modifications on enzyme activity, as quantified by their capacities for phenacetin *O*-deethylation. Notably, all but two variants (Arg34Trp and Leu491Met) showed significantly decreased enzymatic activity, while Leu98Gln, Gly233Arg, Ser380del, Gly454Asp, and Arg457Trp did not exhibit any quantifiable activity. Although we determined *CL_int_* ratios for phenacetin *O*-deethylation, thus outlining the possible influence of these genetic variants on drug therapeutics, further assays are required to discern the substrate specificity of these novel variants and to predict their impact in clinical settings. Additionally, while in silico evaluations are instrumental in preliminary studies, discrepancies between the output of predictive software and the in vitro results further underline the importance of detailed multiapproach studies to avoid data misinterpretation. Current diagnostic tools for the genetic study of the CYP1A2 enzyme are limited to representative allelic variants; our findings further support the need for regional guidelines and greater scrutiny of rare genetic variants to improve therapeutic approaches and discern possible risk, including the influence of environmental factors.

## 5. Conclusions

The enzymatic activities of 21 novel CYP1A2 variants expressed in 293FT cells were characterized by their ability to catalyze phenacetin *O*-deethylation. A majority of the evaluated variants (90%) showed either decreased or negligible enzymatic activity. Therefore, patients harboring these rare genetic variants could be at higher risk of compromised treatment outcomes or experience increased risk from exposure to CYP1A2-inducing environmental factors. The CYP1A2 enzyme has been highly implicated in several pathologies, including increased carcinogenesis, myocardial infarction, and insulin resistance. While our findings provide additional information regarding rare genetic variants found in the Japanese population, additional correlation studies using different clinically relevant CYP1A2 substrates are required to determine the risks associated with treatment and predict potential clinical outcomes. While rare allelic variants can be restricted to specific populations and thus overlooked in traditional diagnostic techniques, their detailed study could improve regional therapeutics by building patient-specific treatment approaches and advancing biomedical research by establishing novel therapeutic targets based on multidisciplinary data regarding CYP1A2 enzyme function.

## Figures and Tables

**Figure 1 jpm-11-00690-f001:**
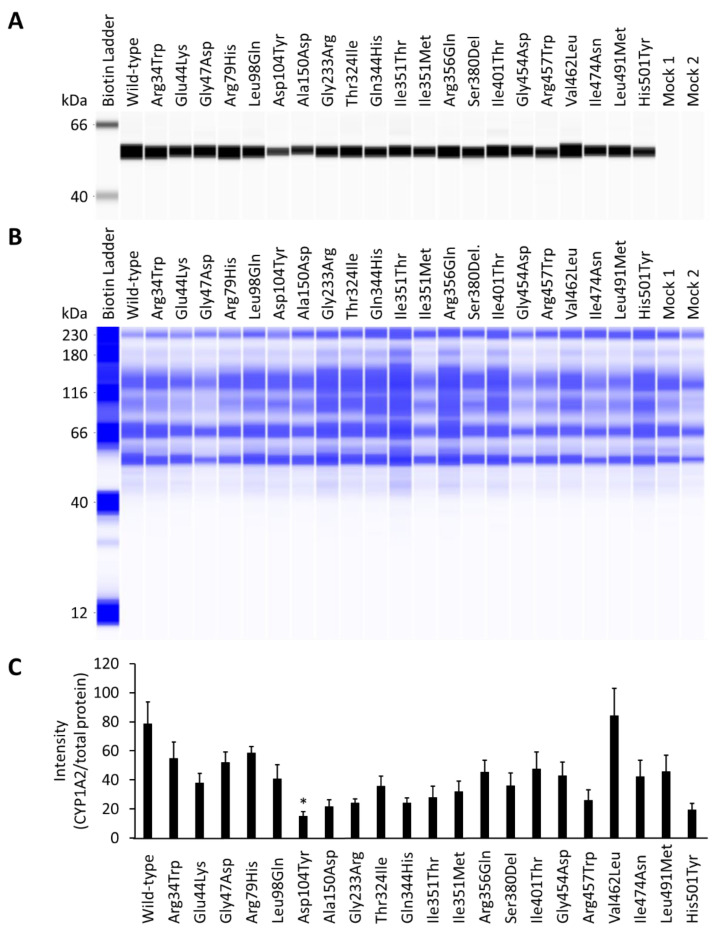
Representative Western blots showing immunoreactive cytochrome P450 1A2 (CYP1A2) proteins (**A**) and total proteins (**B**). Average CYP1A2 intensity was normalized according to total protein content (**C**). All assays and measurements were performed in triplicate using a single microsomal preparation. Mock 1 represents transfection with 10 μg mock plasmid. Mock 2 represents transfection with 9.6 μg mock plasmid, 0.2 μg cytochrome P450 oxidoreductase (CPR) plasmid, and 0.2 μg cytochrome b_5_ plasmid. N.D. represents not determined. * *p* < 0.05 compared with wild-type CYP1A2 by Kruskal–Wallis method.

**Figure 2 jpm-11-00690-f002:**
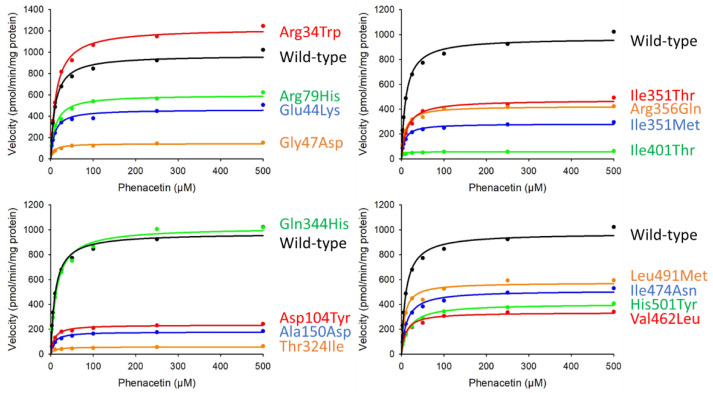
Michaelis–Menten curves for CYP1A2 variants. Determined kinetic parameters of phenacetin *O*-deethylation. All assays and measurements were performed in triplicate using a single microsomal preparation.

**Figure 3 jpm-11-00690-f003:**
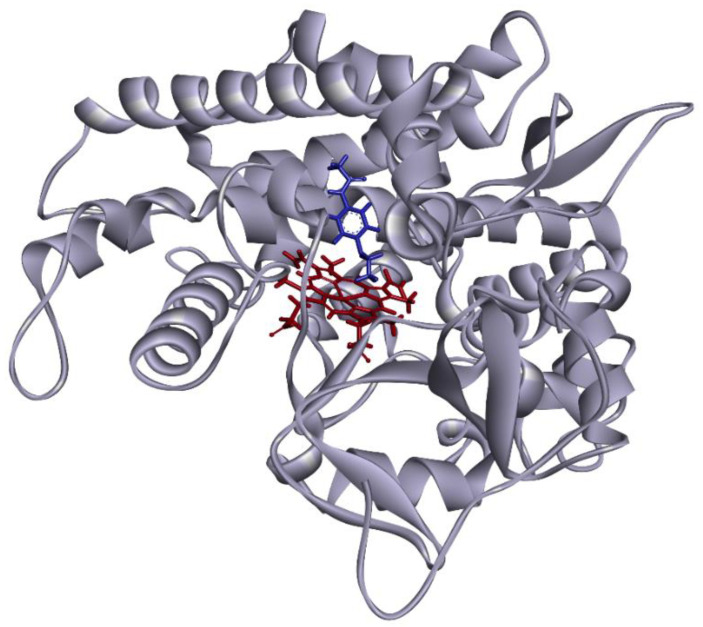
Diagram of the overall structure of CYP1A2. The heme group is shown in red; phenacetin is shown in blue. The distance between the central iron and the metabolic site was calculated as 4.32 Å.

**Figure 4 jpm-11-00690-f004:**
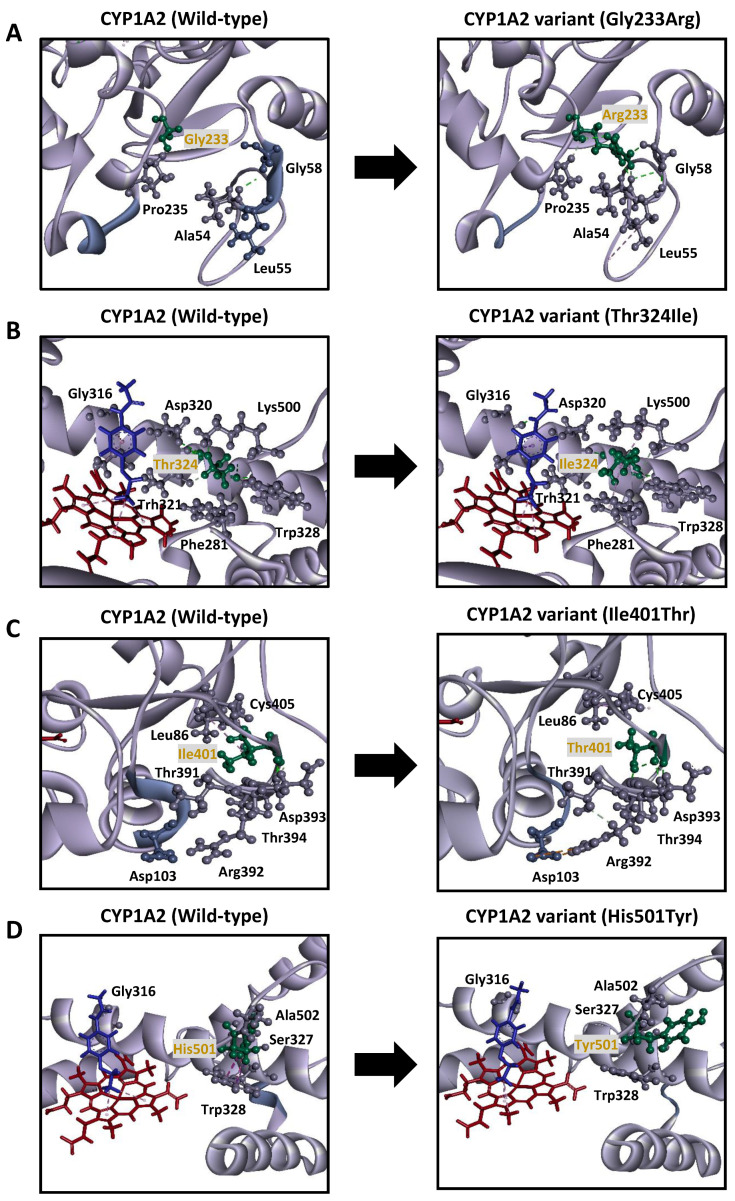
Diagram pairs showing the partial crystal structure of CYP1A2.1 (**left image**) and CYP1A2 variants (**right image**) for Gly233Arg (**A**), Thr324Ile (**B**), Ile401Thr (**C**), and His501Tyr (**D**). The heme group is shown in red; phenacetin is shown in blue, and amino acid substitutions are shown in green. Unwound helical sections are shown in teal blue. Pink line: hydrophobic interactions. Green line: conventional hydrogen bonds. Gray line: carbon-hydrogen bonds. Orange line: electrostatic interaction. Purple line: Pi-Pi T-shaped interaction.

**Table 1 jpm-11-00690-t001:** Novel *CYP1A2* allelic variants characterized in this study.

Nucleotide Mutations	rs Number	Amino Acid Substitutions	Frequency ^a^ (%)	PolyPhen-2	SIFT
100C>T	rs201934979	Arg34Trp	0.01	Benign	Tolerated
130G>A	rs3743482	Glu44Lys	0.10	Benign	Damaging
140G>A	rs777457540	Gly47Asp	0.02	Benign	Damaging
236G>A	rs752037611	Arg79His	0.01	Benign	Tolerated
293T>A	rs773366123	Leu98Gln	0.04	Probably damaging	Damaging
310G>T		Asp104Tyr	0.01	Probably damaging	Damaging
449C>A	rs1477440128	Ala150Asp	0.04	Probably damaging	Tolerated
697G>A	rs201537008	Gly233Arg	0.02	Probably damaging	Damaging
971C>T	rs778097570	Thr324Ile	0.03	Probably damaging	Damaging
1032G>C		Gln344His	0.03	Benign	Tolerated
1052T>C		Ile351Thr	0.01	Benign	Damaging
1053T>G		Ile351Met	0.01	Possibly damaging	Damaging
1067G>A	rs55918015	Arg356Gln	0.02	Benign	Tolerated
1139_1141delCCT	rs757356377	Ser380del	0.01	N.D.	N.D.
1202T>C		Ile401Thr	0.01	Probably damaging	Damaging
1361G>A		Gly454Asp	0.01	Probably damaging	Damaging
1369C>T	rs34151816	Arg457Trp	0.01	Probably damaging	Damaging
1384G>C	rs199528490	Val462Leu	0.04	Benign	Tolerated
1421T>A	rs1300577537	Ile474Asn	0.03	Probably damaging	Damaging
1471C>A		Leu491Met	0.02	Benign	Tolerated
1501C>T		His501Tyr	0.02	Benign	Tolerated

^a^ Allele frequency in 4773 Japanese individuals. N.D. represents not determined.

**Table 2 jpm-11-00690-t002:** The characterization of microsomes prepared from 293FT cells co-expressed with CYP1A2 variants, CPR, and cytochrome b_5_.

Variants	CYP Holoenzyme Content (pmol/mg Protein)	CPR Content (pmol/mg Protein)	Cytochrome b_5_ Content (pmol/mg Protein)	CYP:CPR Ratio	CYP:Cytochrome b_5_ Ratio
Wild-type	145.91 ± 4.11	80.52 ± 16.79	21.11 ± 2.44	1.81	6.91
Arg34Trp	34.50 ± 11.84 *	135.50 ± 45.33	28.34 ± 2.11 ^#^	0.25	1.22
Glu44Lys	41.69 ± 4.62 ***	89.69 ± 20.20	27.14 ± 3.24	0.46	1.54
Gly47Asp	23.21 ± 6.09 ***	98.12 ± 31.17	27.54 ± 2.01	0.24	0.84
Arg79His	21.31 ± 3.02 ***	158.64 ± 24.52	27.09 ± 3.18	0.13	0.79
Leu98Gln	N.D.	146.98 ± 9.95	25.43 ± 1.49	N.D.	N.D.
Asp104Tyr	15.21 ± 1.52 ***	150.87 ± 57.19	26.46 ± 1.69	0.10	0.57
Ala150Asp	20.75 ± 1.41 ***	127.57 ± 47.56	19.28 ± 1.74	0.16	1.08
Gly233Arg	N.D.	109.67 ± 52.40	24.82 ± 2.06	N.D.	N.D.
Thr324Ile	21.89 ± 6.31 ***	103.76 ± 15.09	17.46 ± 3.16	0.21	1.25
Gln344His	18.83 ± 2.93 ***	156.04 ± 4.92	30.98 ± 2.52 ^###^	0.12	0.61
Ile351Thr	36.09 ± 2.85 ***	133.09 ± 12.25	27.72 ± 2.79 ^#^	0.27	1.30
Ile351Met	43.77 ± 1.61 ***	129.07 ± 6.34	17.71 ± 1.17	0.34	2.47
Arg356Gln	88.92 ± 0.88 *	129.54 ± 1.51	25.77 ± 3.52	0.69	3.45
Ser380del	N.D.	140.78 ± 25.66	19.72 ± 2.48	N.D.	N.D.
Ile401Thr	N.D.	95.05 ± 33.43	18.27 ± 3.62	N.D.	N.D.
Gly454Asp	12.88 ± 4.98 ***	120.75 ± 18.54	16.95 ± 1.45	0.11	0.76
Arg457Trp	14.34 ± 2.24 ***	126.46 ± 16.37	17.00 ± 2.12	0.11	0.84
Val462Leu	14.59 ± 5.64 ***	113.72 ± 62.19	23.71 ± 3.57	0.13	0.62
Ile474Asn	33.80 ± 4.70 ***	136.52 ± 28.20	26.53 ± 3.32	0.25	1.27
Leu491Met	144.17 ± 6.56	109.22 ± 35.05	23.54 ± 1.60	1.32	6.13
His501Tyr	43.22 ± 2.31 ***	141.73 ± 9.07	31.52 ± 4.02 ^###^	0.30	1.37

Data represent the means ± SDs of the three independently performed catalytic assays. * *p* < 0.05 and *** *p* < 0.005 compared with wild-type CYP1A2 by Dunnett’s T3 test. ^#^ *p* < 0.05 and ^###^ *p* < 0.005 compared with wild-type CYP1A2 by Dunnett’s *t*-test. N.D. represents not determined. All assays and measurements were performed in triplicate using a single microsomal preparation.

**Table 3 jpm-11-00690-t003:** Kinetic parameters of phenacetin *O*-deethylation by microsomes from 293FT cells expressing wild-type and variant CYP1A2 proteins.

Variants	*K_m_* (μM)	*V_max_* (pmol/min/mg Protein)	*CL_int_* (µL/min/mg Protein) (% of Wild-Type)
Wild-type	10.14 ± 0.68	972.99 ± 39.96	96.10 ± 2.47 (100.00)
Arg34Trp	12.97 ± 0.65	1224.92 ± 48.86	94.51 ± 1.30 (98.35)
Glu44Lys	8.86 ± 0.27	462.54 ± 12.05 *	52.23 ± 2.04 *** (54.35)
Gly47Asp	7.42 ± 0.32	142.46 ± 3.76 **	19.22 ± 0.50 *** (20.00)
Arg79His	11.06 ± 0.74	599.00 ± 17.84 *	54.23 ± 1.99 *** (56.43)
Leu98Gln	N.D.	N.D.	N.D.
Asp104Tyr	7.83 ± 0.43	234.53 ± 5.62 **	30.04 ± 2.06 *** (31.26)
Ala150Asp	7.71 ± 0.27	178.11 ± 3.53 **	23.12 ± 1.13 *** (24.06)
Gly233Arg	N.D.	N.D.	N.D.
Thr324Ile	110.07 ± 2.08	153.36 ± 0.59 **	1.39 ± 0.02 *** (1.45)
Gln344His	13.18 ± 0.30	1019.25 ± 28.36	77.36 ± 2.97 * (80.50)
Ile351Thr	12.92 ± 1.10	473.86 ± 16.59 **	36.93 ± 4.50 *** (38.44)
Ile351Met	7.37 ± 0.58	285.80 ± 16.34 ***	38.85 ± 0.84 *** (40.43)
Arg356Gln	8.03 ± 0.21	423.55 ± 25.17 ***	52.74 ± 1.75 *** (54.88)
Ser380del	N.D.	N.D.	N.D.
Ile401Thr	2.31 ± 0.26	57.05 ± 1.90 **	24.88 ± 2.09 *** (25.89)
Gly454Asp	N.D.	N.D.	N.D.
Arg457Trp	N.D.	N.D.	N.D.
Val462Leu	9.97 ± 1.39	334.57 ± 8.54 **	34.06 ± 5.64 ** (35.44)
Ile474Asn	12.41 ± 1.45	511.48 ± 10.98 *	41.63 ± 5.48 * (43.31)
Leu491Met	6.97 ± 0.41	574.36 ± 9.20 *	82.58 ± 4.18 (85.93)
His501Tyr	16.83 ± 1.29	404.49 ± 14.69 **	24.08 ± 1.13 *** (25.06)

Data represent the means ± SD of the three independently performed catalytic assays. * *p* < 0.05, ** *p* < 0.01, and *** *p* < 0.005 compared with wild-type CYP1A2 by Dunnett’s T3 tests. N.D. represents not determined. All assays and measurements were performed in triplicate using a single microsomal preparation.

## Data Availability

Data available on request due to restrictions, e.g., privacy or ethical.

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
