# Peer review of "Functional Characterization of 21 Rare Allelic CYP1A2 Variants Identified in a Population of 4773 Japanese Individuals by Assessing Phenacetin O-Deethylation"

_jpm, 2021, doi:10.3390/jpm11080690_

Round 1
Reviewer 1 Report
In the manuscript titled “Functional Characterization of 21 Rare Allelic CYP1A2 Variants Identified in a Population of 4773 Japanese Individuals by Assessing Phenacetin O-deethylation” Kumondai et al. have characterized 21 novel CYP1A2 variants that were identified in 4773 Japanese individuals by determining the kinetic parameters of phenacetin O-deethylation. The authors have observed that most of the variants exhibit low or no enzymatic activity when compared to CYP1A2. Docking studies were performed to understand the pharmacokinetics of CYP1A2 variants which revealed destabilization of binding regions and changes in protein folding could be the reason for low activities of variants. The study is useful as it provides information to improve personalized therapeutics that involve CYP1A2.
Author Response
We thank the reviewer for these positive comments and careful review.
Reviewer 2 Report
"Functional Characterization of 21 Rare Allelic CYP1A2 Variants Identified in a Population of 4773 Japanese Individuals by As-sessing Phenacetin O-deethylation" is an interesting clinical study. Below are my comments :
- The western blot analysis is important for this manuscript hence it should be included as main figure. Also, the method for determining total protein content is not clear. "A total protein assay was performed to normalize each signal using 100 ng of microsomes, as per the manufacturer’s instructions." : This has no reference.
- The authors should mention full form for all the acronyms used in this study. This is very critical for the understanding of the readers from diverse backgrounds.
Author Response
We thank the reviewer for the valuable comments, which helped us improve our manuscript. As the reviewer suggested, we have added the western blot analysis results in this manuscript as the main Figure. We have added the following sentence in the Materials and Methods section regarding the methods for determining total protein content. As per the reviewer’s suggestion, we have mentioned a complete form for all the acronyms used in this study, such as Km; Michaelis constant, Vmax; maximum velocity, and CLint; intrinsic clearance.
-Page 3, Western Blotting section
“A total protein assay was performed to normalize each signal using 100 ng of microsomes, as per the manufacturer’s instructions (https://www.proteinsimple.com/simple_western_size-based_total_protein_assays.html).”
Reviewer 3 Report
The article by Kumondai et al investigated the CYP1A2 variants within a sample population of about 5K people in order to characterize their functionality with regard to phenacetin treatment.
The measurements and the modeling results identify molecular level insights into how certain variants exhibit benign vs damaging enzymatic activity via the changes incurred from protein folding processes. In general, the study is convincing in it's assessment of how individuals with the CYP1A2 variants could be at a higher risk of specific treatment outcomes.
This work should be read by a wider scientific community.
Author Response
We thank the reviewer for these constructive comments.